# On Two Non-Ergodic Reversible Cellular Automata, One Classical, the Other Quantum [note 1]

**DOI:** 10.3390/e25050739

**Published:** 2023-04-30

**Authors:** Tomaž Prosen

**Affiliations:** Faculty of Mathematics and Physics, University of Ljubljana, Jadranska 19, SI-1000 Ljubljana, Slovenia; tomaz.prosen@fmf.uni-lj.si

**Keywords:** cellular automata, interacting dynamics, ergodic theory, ergodicity breaking, integrability, conservation laws

## Abstract

We propose and discuss two variants of kinetic particle models—cellular automata in 1 + 1 dimensions—that have some appeal due to their simplicity and intriguing properties, which could warrant further research and applications. The first model is a deterministic and reversible automaton describing two species of quasiparticles: stable massless *matter* particles moving with velocity ±1 and unstable standing (zero velocity) *field* particles. We discuss two distinct continuity equations for three conserved charges of the model. While the first two charges and the corresponding currents have support of three lattice sites and represent a lattice analogue of the conserved energy–momentum tensor, we find an additional conserved charge and current with support of nine sites, implying non-ergodic behaviour and potentially signalling integrability of the model with a highly nested R-matrix structure. The second model represents a quantum (or stochastic) deformation of a recently introduced and studied charged hardpoint lattice gas, where particles of different binary charge (±1) and binary velocity (±1) can nontrivially mix upon elastic collisional scattering. We show that while the unitary evolution rule of this model does not satisfy the full Yang–Baxter equation, it still satisfies an intriguing related identity which gives birth to an infinite set of local conserved operators, the so-called glider operators.

## 1. Introduction

This article is dedicated to my friend and mentor, Giulio Casati. For that reason I will also take the liberty to write the paper from a very personal perspective. One of the most important things that Giulio taught me, was to passionately appreciate extremely simple models of dynamics. Constructing and solving simple models of (lately, most often many-body) dynamics with carefully chosen physical properties has thus become my personal obsession throughout my career. The other thing which I owe Giulio is a pragmatic but deep appreciation of ergodic theory. Together we have been responsible for a few endeavours in experimental mathematics, which sometimes aroused the attention of experts. For instance, we pointed out intriguing ergodic properties of classical and quantum polygonal billiards and related non-hyperbolic and non-integrable dynamical systems [1,2,3,4].

Crudely speaking, ergodic theory [5] divides dynamical systems into ergodic and non-ergodic ones. The former are characterized by the property that time averages of (relevant) observables can be computed in terms of an ensemble average over the entire space of states (phase space), while the latter exhibit memory of the initial condition in typical (say, physically relevant) observables. Additionally, ergodic dynamics can have various degrees of dynamical complexity or chaos, while non-ergodic ones can conform to some algebraic tools of exact solvability, or integrability. However, it is not clear to what extent the breaking of ergodicity is connected to any form of integrability, as it can precisely be defined in terms of Lax pairs (in classical, deterministic systems) or the Yang–Baxter equation (in a quantum or stochastic setting). In the physics literature, these questions have been extensively discussed recently in the context of many-body dynamics, where the mathematics of ergodic theory is much less developed, say on the lattice and having local interactions, both in the classical and quantum realms (see, e.g., some recent reviews [6,7,8,9]). Besides integrability, many different forms of ergodicity breaking have been suggested, such as phase-space or Hilbert-space fragmentation, due to various forms of kinetic constraints, the so-called many-body localization due to static disorder, many-body scarring due to hidden weakly broken non-abelian symmetries, etc.

In this paper, we propose two very simple many-body locally interacting dynamical systems defined on a discrete 1 + 1 space-time lattice. The first model, which we call a matter–field automaton, for reasons that will hopefully appear clear to the reader, is an example of a deterministic particle kinematics with two distinct types of degrees of freedom (matter and field), which is non-ergodic for a nontrivial reason: existence of a highly nontrivial but local conserved charge. Inspired by brute-force empirical computer-algebra calculations, we suggest that the model may be integrable, but if it is, the corresponding R-matrix should be highly nontrivial. The second model, which is a quantum deformation of the previously studied charged hardpoint lattice gas [10,11,12], obeys a remarkable reduced Yang–Baxter-like identity for a non-abelian spectral parameter. This again ensures manifest ergodicity breaking in the model through an existence of an infinite set of local translationally invariant conserved operators, the so-called gliders.

## 2. Matter–Field Automaton

### 2.1. Definition of the Automaton

Let us define a deterministic reversible cellular automaton on ${\left\{0,1\right\}}^{Z}∋\underline{s}$ as follows. Even lattice sites correspond to the *matter* variable $s_{2x}\in \left\{0,1\right\}$, while odd lattice sites correspond to the *field* variable $s_{2x+1}\in \left\{0,1\right\}$, x∈Z. The instantaneous system configuration is thus specified by an infinite binary sequence $\underline{s}$, on which we specify deterministic dynamics.

The dynamics is defined in a staggered fashion, in even and odd time layers. Specifically, for so-called *even* time steps we update the matter–field–matter triples $\left(s_{4x},s_{4x+1},s_{4x+2}\right)$ for all x∈Z with a deterministic, reversible rule $\left(ss^{\prime }s^{\prime \prime }\right)⟶\left(rr^{\prime }r^{\prime \prime }\right)$, while for *odd* time-steps, the triples $\left(s_{4x-2},s_{4x-1},s_{4x}\right)$, are updated by the same rule. The rule is specified as
(1)$$\begin{array}{cc} & (000)⟷(000),\\ & (001)⟷(100),\\ & (011)⟷(110),\\ & (101)⟷(010),\\ & (111)⟷(111), \end{array}$$
or, graphically and perhaps more intuitively, by the following diagrams

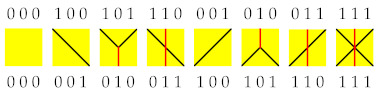

which indicate mappings of all eight lower matter–field–matter triples to corresponding upper matter–field–matter triples, or vice versa due to time-reversal symmetry. The black and red lines thus represent the worldlines of matter and field particles, respectively. In Figure 1, we show an example of a piece of a many-body trajectory $\left\{s_{x}^{t};x,t\in Z\right\}$, where time *t* runs in the vertical direction (upwards). The construction of the trajectory can be understood as a checkerboard tiling of the rule’s yellow squares, which is uniquely prescribed by the lowest two lines of tiles (the initial condition).

Note that this dynamical system has a rich set of three independent $Z^{2}$ symmetries, specifically C, P, and T:“Charge” conjugation C, s→1−s, i.e., if $s_{x}^{t}$ is a valid trajectory, then $1-s_{x}^{t}$ is a valid trajectory,Parity P, x→−x, i.e., if $s_{x}^{t}$ is a valid trajectory, then $s_{-x}^{t}$ is a valid trajectory,Time reversal T, t→−t, i.e., if $s_{x}^{t}$ is a valid trajectory, then $s_{x}^{-t}$ is a valid trajectory.

In Figure 2, Figure 3, Figure 4, Figure 5 and Figure 6, we depict some interesting characteristic trajectories of the matter–field automaton. In Figure 2, we prepare a maximum entropy initial state in a finite box of size *L*, where both field and matter are sampled randomly and uniformly in the set {0,1}, and plot an emerging deterministic trajectory. Since the initial condition vanishes outside the box, $s_{x}^{t=0}=0$, for x<0 or x>L, we see a slow ‘evaporation’ of matter particles from the edges of the box. In Figure 3, we plot a similar trajectory, where initially we have only matter (at maximum entropy) and no field, while in Figure 4 we plot the reverse situation, where initially there is only the field at maximum entropy and no matter. The fact that spatio-temporal patterns of these trajectories seem qualitatively distinct depending on the initial conditions, and in particular the relative ratio of field versus matter drastically differs, suggests that the dynamics may be non-ergodic. We shall establish this more precisely below by constructing nontrivial algebraic conservation laws. While the previous plots represent cartoons of equilibrium dynamics, we plot a typical trajectory representing far-from-equilibrium dynamics in Figure 5: a head-on collision of two streams of matter (with no field in the initial data), which experience a nontrivial scattering process via formation of a complex field pattern, finally resulting in two oppositely moving scattered streams of matter.

Finally, we demonstrate that the matter–field system can be understood as a very general automaton which can also mimic the dynamics of the popular Rule 54 reversible cellular automaton [13] for a dynamically closed (invariant) subset of initial configurations. More generally, we can show it can simulate dynamics for classes of negative-length hard-rod systems with commensurate initial data. For instance, if we take initial condition $\underline{s}$ where only every sixth site can be occupied, i.e., $s_{x}\equiv 0$ for x≠0(mod6), then one can show that no field worldline can be crossed by another matter particle, and hence each field particle decays at the immediate next time step (the field is maximally short-lived). This means that scattering of matter particles always experiences a fixed phase-shift, which makes the dynamics qualitatively similar (yet not precisely equivalent) to Rule 54 on a certain reduced lattice (see Figure 6 for a demonstration). In Rule 54 dynamics, for instance, the minimal distance between movers/kinks (=2) is twice the time-tag in kink-scattering (=1), while in this case, the minimal distance between (left/right) movers (=6) is three times the scattering time-lag (=2), i.e., free field lifetime. A similar commensurate negative-length hard-rod dynamics is obtained for the subset of initial data $\left\{\underline{s}\right\}$ of the form $s_{x^{\prime }}\equiv 0$, $x^{\prime }\neq 0\;\left(\mathrm{mod}\;2\left(2k+1\right)\right)$, for any k=1,2…, where the ratio between the minimal distance between parallel kinks to scattering-lag is 2k+1.

### 2.2. Time Evolution over Abelian Algebra of Local Observables

Let us now reinterpret the dynamics of the matter–field automaton in terms of an abelian $C^{*}$-algebraic dynamical system, and then study its conservation laws. We begin by defining a quasilocal algebra of observables, i.e., real functions over space of configurations ${\left\{0,1\right\}}^{Z}$. For more details on a fully analogous but more detailed construction in a related model of Rule 54 reversible cellular automaton, see ref. [14]. Let ${\left[0\right]}_{2x}$, ${\left[1\right]}_{2x}$ define a local basis of observables, checking if $s_{2x}=0$ (or 1) for the matter degrees of freedom, and correspondingly ${\left[0\right]}_{2x+1},{\left[1\right]}_{2x+1}$ for the field:(2)$${\left[r\right]}_{x}\left(\underline{s}\right)=\delta_{r,s_{x}}.$$
We have ${\left[0\right]}_{x}+{\left[1\right]}_{x}=1$ (unit element in the algebra of observables), and denote ${\left[•\right]}_{x}={\left[1\right]}_{x}-{\left[0\right]}_{x}$, for *x* either even or odd. We note that ${\left[0\right]}_{x},{\left[1\right]}_{x}$ (or $1,{\left[•\right]}_{x}$) generate the entire (quasi)local algebra under multiplication and addition (and closure). We denote local observables with larger support as ${\left[s_{0},s_{1},\ldots ,s_{r}\right]}_{x}={\left[s_{0}\right]}_{x}{\left[s_{1}\right]}_{x+1}\cdots {\left[s_{r}\right]}_{x+r}$.

The fundamental three site operator—local propagator—in the algebra of observables implementing the rules
(3)$$\begin{array}{cc} & {\left[000\right]}_{2x}⟷{\left[000\right]}_{2x},\\ & {\left[001\right]}_{2x}⟷{\left[100\right]}_{2x},\\ & {\left[011\right]}_{2x}⟷{\left[110\right]}_{2x},\\ & {\left[101\right]}_{2x}⟷{\left[010\right]}_{2x},\\ & {\left[111\right]}_{2x}⟷{\left[111\right]}_{2x}, \end{array}$$
is encoded in terms of the 8×8 permutation matrix
(4)$$Y=\left(\begin{array}{cccccccc} 1 & 0 & 0 & 0 & 0 & 0 & 0 & 0\\ 0 & 0 & 0 & 0 & 1 & 0 & 0 & 0\\ 0 & 0 & 0 & 0 & 0 & 1 & 0 & 0\\ 0 & 0 & 0 & 0 & 0 & 0 & 1 & 0\\ 0 & 1 & 0 & 0 & 0 & 0 & 0 & 0\\ 0 & 0 & 1 & 0 & 0 & 0 & 0 & 0\\ 0 & 0 & 0 & 1 & 0 & 0 & 0 & 0\\ 0 & 0 & 0 & 0 & 0 & 0 & 0 & 1 \end{array}\right)\;.$$
The full time step of dynamical automorphism on the algebra of observables $a^{t}\left(\underline{s}\right)\equiv a\left({\underline{s}}^{-t}\right)$,
(5)$$a^{2(t+1)}=Ua^{2t},\;a^{2t+1}=U_{e}a^{2t},\;a^{2t+2}=U_{o}a^{2t+1},$$
is constructed as follows
(6)$$U=U_{o}U_{e}\;,$$
where even and odd time steps are generated as
(7)$$\begin{array}{c} U_{e}=\prod_{x}Y_{4x,4x+1,4x+2},\;U_{o}=\prod_{x}Y_{4x-2,4x-1,4x}\;, \end{array}$$
where three indices denote the positions in the string, or in the tensor product of local algebras (${⨂}_{x}\mathrm{span}\left\{{\left[0\right]}_{x},{\left[1\right]}_{x}\right\}$), where the three site operator (4) acts nontrivially.

### 2.3. Conservation Laws

Let $a_{\left[x,y\right]}$ be any local observable supported in the interval of sites [x,y]⊂Z. We define a shift automorphism Π
(8)$$\Pi a_{\left[x,y\right]}=a_{\left[x+2,y+2\right]}\;,$$
which shifts an observable by ‘one lattice unit’, so that the identity of matter/field sites is preserved. Note that
(9)$$\Pi U^{e}=U^{o}\Pi ,\;\Pi U^{o}=U^{e}\Pi ,\;\left[\Pi^{2},U^{e}\right]=0,\;\left[\Pi^{2},U^{o}\right]=0.$$
A local observable *q* is called a conserved charge if, together with an appropriate conserved current *p*, it satisfies a conservation law (space–time discrete continuity equation):(10)$$U_{e}q-U_{o}q+\Pi p-\Pi^{-1}p=0,$$
where we note that
(11)$$U_{o}^{2}=U_{e}^{2}=1\;,$$
since $Y^{2}=1$. Then, extensive local observable is globally conserved
(12)$$Q=\sum_{x\in Z}q_{x},\;q_{x}:=\Pi^{2x}q,\;UQ=Q.$$
Importantly, it holds in general that if (q,p) is a local conservation law, then (Πq,−Πp) is a conservation law as well. This can be proven by multiplying Equation (10) by Π and using (9).

Limiting to local observables on clusters of five sites (beginning and ending with a matter site), covering one ’unit cell’ (periodicity block of 4 sites) of the automaton, we find (using exact computer algebra) exactly two inequivalent conservation laws
(13)$$\begin{array}{ccc} q_{x}^{\left(1\right)} & = & {\left[•\right]}_{4x}-{\left[•\right]}_{4x+2}, \end{array}$$
(14)$$\begin{array}{ccc} p_{x}^{\left(1\right)} & = & \frac{1}{2}{\left[•\right]}_{4x}+\frac{1}{2}{\left[•\right]}_{4x+1}+\frac{1}{2}{\left[•\right]}_{4x+2}+\frac{1}{2}{\left[•\right]}_{4x}{\left[•\right]}_{4x+1}{\left[•\right]}_{4x+2}, \end{array}$$
(15)$$\begin{array}{ccc} q_{x}^{\left(2\right)} & = & {\left[•\right]}_{4x+1}+{\left[•\right]}_{4x+2}+{\left[•\right]}_{4x+3}, \end{array}$$
(16)$$\begin{array}{ccc} p_{x}^{\left(2\right)} & = & -{\left[•\right]}_{4x+2}, \end{array}$$
with local densities and current supported on up to three consecutive sites. $q_{x}^{\left(1\right)}$ can be interpreted as the *momentum* density, while $q_{x}^{\left(2\right)}$ is the analog of the *energy* density. Indeed, writing $P=Q^{\left(1\right)}=\sum_{x}q_{x}^{\left(1\right)}$, $E=Q^{\left(2\right)}+\Pi Q^{\left(2\right)}=\sum_{x}\left(q_{x}^{\left(2\right)}+\Pi q_{x}^{\left(2\right)}\right)$ we have intuitive expressions for the total momentum and total energy of a configuration:(17)$$P\left(\underline{s}\right)=\sum_{x}{\left(-1\right)}^{x}s_{2x}\;,\;E\left(\underline{s}\right)=\sum_{x}\left(s_{2x}+2s_{2x+1}\right)\;,$$
meaning that the field particle carries no momentum and has an energy content (rest energy) equal to the energy of two matter particles. Note that a pair (P,E) is equivalent to a pair $\left(Q^{\left(1\right)},Q^{\left(2\right)}\right)$, since $\Pi Q^{\left(1\right)}=\sum_{x}\Pi q_{x}^{\left(1\right)}=-P$ and $Q^{\left(2\right)}-\Pi Q^{\left(2\right)}=\sum_{x}\left(q_{x}^{\left(2\right)}-\Pi q_{x}^{\left(2\right)}\right)=-\sum_{x}q_{x}^{\left(1\right)}=-P$.

If we then increase the support size to clusters of nine (2×4+1) consecutive sites, we obtain precisely one new additional conservation law:

(using compact notation: [∘]≡1, ${\left[r_{0}r_{1}r_{2}\ldots \right]}_{4x}\equiv \left[r_{0}r_{1}r_{2}\ldots \right]$)
(18)$$\begin{array}{cc} & \;\;\;\;\;\;\;\;\;\;\;\;\;\;q^{\left(3\right)}=2\left[\circ \circ \circ \circ \circ \circ •\circ •\right]-2\left[\circ \circ \circ \circ \circ \circ ••\circ \right]-4\left[\circ \circ \circ \circ \circ •\circ \circ •\right]-2\left[\circ \circ \circ \circ \circ ••\circ \circ \right]\\ & \;\;\;\;\;\;\;\;\;\;\;\;\;\;\;\;\;\;\;\;\;\;\;\;+2[\circ \circ \circ \circ \circ ••••]-8[\circ \circ \circ \circ •\circ \circ \circ •]-4[\circ \circ \circ \circ •\circ \circ •\circ ]+2[\circ \circ \circ \circ •\circ •\circ \circ ]\\ & \;\;\;\;\;\;\;\;\;\;\;\;\;\;\;\;\;\;\;\;\;\;\;\;-2[\circ \circ \circ \circ •\circ •••]+4[\circ \circ \circ \circ ••\circ ••]-2[\circ \circ \circ \circ •••\circ •]+2[\circ \circ \circ \circ ••••\circ ]\\ & \;\;\;\;\;\;\;\;\;\;\;\;\;\;\;\;\;\;\;\;\;\;\;\;-2[\circ \circ \circ •\circ \circ \circ \circ •]-2[\circ \circ \circ •\circ \circ \circ •\circ ]+2[\circ \circ \circ •\circ \circ •\circ \circ ]-2[\circ \circ \circ •\circ \circ •••]\\ & \;\;\;\;\;\;\;\;\;\;\;\;\;\;\;\;\;\;\;\;\;\;\;\;-4[\circ \circ \circ •\circ •\circ \circ \circ ]+2[\circ \circ \circ •••\circ \circ •]+2[\circ \circ \circ •••\circ •\circ ]-2[\circ \circ \circ ••••\circ \circ ]\\ & \;\;\;\;\;\;\;\;\;\;\;\;\;\;\;\;\;\;\;\;\;\;\;\;+2[\circ \circ \circ ••••••]+[\circ \circ •\circ \circ \circ \circ \circ •]+[\circ \circ •\circ \circ \circ \circ •\circ ]-[\circ \circ •\circ \circ \circ •\circ \circ ]\\ & \;\;\;\;\;\;\;\;\;\;\;\;\;\;\;\;\;\;\;\;\;\;\;\;+[\circ \circ •\circ \circ \circ •••]+2[\circ \circ •\circ \circ •\circ \circ \circ ]-[\circ \circ •\circ ••\circ \circ •]-[\circ \circ •\circ ••\circ •\circ ]\\ & \;\;\;\;\;\;\;\;\;\;\;\;\;\;\;\;\;\;\;\;\;\;\;\;+[\circ \circ •\circ •••\circ \circ ]-[\circ \circ •\circ •••••]+[\circ \circ ••\circ •\circ \circ •]+[\circ \circ ••\circ •\circ •\circ ]\\ & \;\;\;\;\;\;\;\;\;\;\;\;\;\;\;\;\;\;\;\;\;\;\;\;-[\circ \circ ••\circ ••\circ \circ ]+[\circ \circ ••\circ ••••]-[\circ \circ •••\circ \circ \circ •]-[\circ \circ •••\circ \circ •\circ ]\\ & \;\;\;\;\;\;\;\;\;\;\;\;\;\;\;\;\;\;\;\;\;\;\;\;+[\circ \circ •••\circ •\circ \circ ]-[\circ \circ •••\circ •••]-2[\circ \circ ••••\circ \circ \circ ]-[\circ •\circ \circ \circ \circ \circ \circ •]\\ & \;\;\;\;\;\;\;\;\;\;\;\;\;\;\;\;\;\;\;\;\;\;\;\;-[\circ •\circ \circ \circ \circ \circ •\circ ]+[\circ •\circ \circ \circ \circ •\circ \circ ]-[\circ •\circ \circ \circ \circ •••]-2[\circ •\circ \circ \circ •\circ \circ \circ ]\\ & \;\;\;\;\;\;\;\;\;\;\;\;\;\;\;\;\;\;\;\;\;\;\;\;+[\circ •\circ \circ ••\circ \circ •]+[\circ •\circ \circ ••\circ •\circ ]-[\circ •\circ \circ •••\circ \circ ]+[\circ •\circ \circ •••••]\\ & \;\;\;\;\;\;\;\;\;\;\;\;\;\;\;\;\;\;\;\;\;\;\;\;-[\circ •\circ •\circ •\circ \circ •]-[\circ •\circ •\circ •\circ •\circ ]+[\circ •\circ •\circ ••\circ \circ ]-[\circ •\circ •\circ ••••]\\ & \;\;\;\;\;\;\;\;\;\;\;\;\;\;\;\;\;\;\;\;\;\;\;\;+[\circ •\circ ••\circ \circ \circ •]+[\circ •\circ ••\circ \circ •\circ ]-[\circ •\circ ••\circ •\circ \circ ]+[\circ •\circ ••\circ •••]\\ & \;\;\;\;\;\;\;\;\;\;\;\;\;\;\;\;\;\;\;\;\;\;\;\;+2[\circ •\circ •••\circ \circ \circ ]-[•\circ \circ \circ \circ \circ \circ \circ •]-[•\circ \circ \circ \circ \circ \circ •\circ ]+[•\circ \circ \circ \circ \circ •\circ \circ ]\\ & \;\;\;\;\;\;\;\;\;\;\;\;\;\;\;\;\;\;\;\;\;\;\;\;-[•\circ \circ \circ \circ \circ •••]-2[•\circ \circ \circ \circ •\circ \circ \circ ]+[•\circ \circ \circ ••\circ \circ •]+[•\circ \circ \circ ••\circ •\circ ]\\ & \;\;\;\;\;\;\;\;\;\;\;\;\;\;\;\;\;\;\;\;\;\;\;\;-[•\circ \circ \circ •••\circ \circ ]+[•\circ \circ \circ •••••]-[•\circ \circ •\circ •\circ \circ •]-[•\circ \circ •\circ •\circ •\circ ]\\ & \;\;\;\;\;\;\;\;\;\;\;\;\;\;\;\;\;\;\;\;\;\;\;\;+[•\circ \circ •\circ ••\circ \circ ]-[•\circ \circ •\circ ••••]+[•\circ \circ ••\circ \circ \circ •]+[•\circ \circ ••\circ \circ •\circ ]\\ & \;\;\;\;\;\;\;\;\;\;\;\;\;\;\;\;\;\;\;\;\;\;\;\;-[•\circ \circ ••\circ •\circ \circ ]+[•\circ \circ ••\circ •••]+2[•\circ \circ •••\circ \circ \circ ]-[•••\circ \circ \circ \circ \circ •]\\ & \;\;\;\;\;\;\;\;\;\;\;\;\;\;\;\;\;\;\;\;\;\;\;\;-[•••\circ \circ \circ \circ •\circ ]+[•••\circ \circ \circ •\circ \circ ]-[•••\circ \circ \circ •••]-2[•••\circ \circ •\circ \circ \circ ]\\ & \;\;\;\;\;\;\;\;\;\;\;\;\;\;\;\;\;\;\;\;\;\;\;\;+[•••\circ ••\circ \circ •]+[•••\circ ••\circ •\circ ]-[•••\circ •••\circ \circ ]+[•••\circ •••••]\\ & \;\;\;\;\;\;\;\;\;\;\;\;\;\;\;\;\;\;\;\;\;\;\;\;-[••••\circ •\circ \circ •]-[••••\circ •\circ •\circ ]+[••••\circ ••\circ \circ ]-[••••\circ ••••]\\ & \;\;\;\;\;\;\;\;\;\;\;\;\;\;\;\;\;\;\;\;\;\;\;\;+[•••••\circ \circ \circ •]+[•••••\circ \circ •\circ ]-[•••••\circ •\circ \circ ]+[•••••\circ •••]\\ & \;\;\;\;\;\;\;\;\;\;\;\;\;\;\;\;\;\;\;\;\;\;\;\;+2[••••••\circ \circ \circ ] \end{array}$$
with the current
(19)$$\begin{array}{cc} & \;\;\;\;\;\;\;\;\;\;\;\;p^{\left(3\right)}=2\left[\circ \circ \circ \circ \circ ••\circ \circ \right]-4\left[\circ \circ \circ \circ •\circ \circ \circ •\right]-4\left[\circ \circ \circ \circ •\circ \circ •\circ \right]+2\left[\circ \circ \circ \circ •\circ •\circ \circ \right]\\ & \;\;\;\;\;\;\;\;\;\;\;\;\;\;\;\;\;\;\;\;\;\;\;\;-4[\circ \circ \circ \circ •\circ •••]-2[\circ \circ \circ •\circ \circ \circ \circ •]-2[\circ \circ \circ •\circ \circ \circ •\circ ]-2[\circ \circ \circ •\circ \circ •••]\\ & \;\;\;\;\;\;\;\;\;\;\;\;\;\;\;\;\;\;\;\;\;\;\;\;+2[\circ \circ \circ •••\circ \circ •]+2[\circ \circ \circ •••\circ •\circ ]+2[\circ \circ \circ ••••••]+[\circ \circ •\circ \circ \circ \circ \circ •]\\ & \;\;\;\;\;\;\;\;\;\;\;\;\;\;\;\;\;\;\;\;\;\;\;\;+[\circ \circ •\circ \circ \circ \circ •\circ ]+[\circ \circ •\circ \circ \circ •••]-[\circ \circ •\circ ••\circ \circ •]-[\circ \circ •\circ ••\circ •\circ ]\\ & \;\;\;\;\;\;\;\;\;\;\;\;\;\;\;\;\;\;\;\;\;\;\;\;-[\circ \circ •\circ •••••]+[\circ \circ ••\circ •\circ \circ •]+[\circ \circ ••\circ •\circ •\circ ]+[\circ \circ ••\circ ••••]\\ & \;\;\;\;\;\;\;\;\;\;\;\;\;\;\;\;\;\;\;\;\;\;\;\;-[\circ \circ •••\circ \circ \circ •]-[\circ \circ •••\circ \circ •\circ ]-[\circ \circ •••\circ •••]-[\circ •\circ \circ \circ \circ \circ \circ •]\\ & \;\;\;\;\;\;\;\;\;\;\;\;\;\;\;\;\;\;\;\;\;\;\;\;-[\circ •\circ \circ \circ \circ \circ •\circ ]-[\circ •\circ \circ \circ \circ •••]+[\circ •\circ \circ ••\circ \circ •]+[\circ •\circ \circ ••\circ •\circ ]\\ & \;\;\;\;\;\;\;\;\;\;\;\;\;\;\;\;\;\;\;\;\;\;\;\;+[\circ •\circ \circ •••••]-[\circ •\circ •\circ •\circ \circ •]-[\circ •\circ •\circ •\circ •\circ ]-[\circ •\circ •\circ ••••]\\ & \;\;\;\;\;\;\;\;\;\;\;\;\;\;\;\;\;\;\;\;\;\;\;\;+[\circ •\circ ••\circ \circ \circ •]+[\circ •\circ ••\circ \circ •\circ ]+[\circ •\circ ••\circ •••]-[•\circ \circ \circ \circ \circ \circ \circ •]\\ & \;\;\;\;\;\;\;\;\;\;\;\;\;\;\;\;\;\;\;\;\;\;\;\;-[•\circ \circ \circ \circ \circ \circ •\circ ]-[•\circ \circ \circ \circ \circ •••]+[•\circ \circ \circ ••\circ \circ •]+[•\circ \circ \circ ••\circ •\circ ]\\ & \;\;\;\;\;\;\;\;\;\;\;\;\;\;\;\;\;\;\;\;\;\;\;\;+[•\circ \circ \circ •••••]-[•\circ \circ •\circ •\circ \circ •]-[•\circ \circ •\circ •\circ •\circ ]-[•\circ \circ •\circ ••••]\\ & \;\;\;\;\;\;\;\;\;\;\;\;\;\;\;\;\;\;\;\;\;\;\;\;+[•\circ \circ ••\circ \circ \circ •]+[•\circ \circ ••\circ \circ •\circ ]+[•\circ \circ ••\circ •••]-[•••\circ \circ \circ \circ \circ •]\\ & \;\;\;\;\;\;\;\;\;\;\;\;\;\;\;\;\;\;\;\;\;\;\;\;-[•••\circ \circ \circ \circ •\circ ]-[•••\circ \circ \circ •••]+[•••\circ ••\circ \circ •]+[•••\circ ••\circ •\circ ]\\ & \;\;\;\;\;\;\;\;\;\;\;\;\;\;\;\;\;\;\;\;\;\;\;\;+[•••\circ •••••]-[••••\circ •\circ \circ •]-[••••\circ •\circ •\circ ]-[••••\circ ••••]\\ & \;\;\;\;\;\;\;\;\;\;\;\;\;\;\;\;\;\;\;\;\;\;\;\;+[•••••\circ \circ \circ •]+[•••••\circ \circ •\circ ]+[•••••\circ •••]. \end{array}$$
If we furthermore increase this to a 3-cell cluster of 13=3×4+1 consecutive sites, we obtain only one additional conservation law which, however, is trivially related to the previous one:(20)$$q_{x}^{\left(4\right)}=\Pi q_{x}^{\left(3\right)},\;p_{x}^{\left(4\right)}=-\Pi p_{x}^{\left(3\right)}.$$
Scanning for the existence of higher conservation laws with densities supported over four or more unit cells seems impossible with brute-force computer algebra.

Furthermore, note that the conservation law property (10) is invariant under local space–time discrete gauge transformation, i.e., taking any local observable *a*
(21)$$\begin{array}{cc} q & ⟶q+\Pi a-\Pi^{-1}a, \end{array}$$
(22)$$\begin{array}{cc} p & ⟶p-U_{e}a+U_{o}a. \end{array}$$
In the explicit expressions listed above, the gauge has been fixed by right-alignment of the densities.

One might expect that matter–field automaton hosts an infinite tower of local conserved quantities of increasing support size, however, these may be very difficult to find empirically. This hypothesis would suggest also that the matter–field automaton may be a completely integrable system, in a similar spirit as Rule 54 [15], as it also displays similar negative-length hard-rod dynamics for a class of initial data, but preliminary attempts to find a Yang–Baxter or Lax structure failed. The other option is that the matter–field system has only a few, perhaps finitely many (or an incomplete set) conserved local charges, and would then represent a paradigmatic theory between integrable and ergodic dynamics.

Deterministic cellular automata on discrete phase space (space of configurations) often admit continuous, single (or few) parameter modifications, which render the models quantum or stochastic (depending on the type of deformation which turns a deterministic evolution to a unitary or Markov map). For instance, in the definition of the matter–field automaton, one may in each step allow for the triple (010) to remain (010) with probability α and to map to (101) with probability 1−α. Similarly, (101) remains in (101) with probability β and maps to (010) with probability 1−β. All the definitions of the previous section remain valid, except that the permutation matrix *Y* is now replaced by a stochastic (Markov) matrix:(23)$$Y_{\mathrm{stoch}}=\left(\begin{array}{cccccccc} 1 & 0 & 0 & 0 & 0 & 0 & 0 & 0\\ 0 & 0 & 0 & 0 & 1 & 0 & 0 & 0\\ 0 & 0 & \alpha & 0 & 0 & 1-\beta & 0 & 0\\ 0 & 0 & 0 & 0 & 0 & 0 & 1 & 0\\ 0 & 1 & 0 & 0 & 0 & 0 & 0 & 0\\ 0 & 0 & 1-\alpha & 0 & 0 & \beta & 0 & 0\\ 0 & 0 & 0 & 1 & 0 & 0 & 0 & 0\\ 0 & 0 & 0 & 0 & 0 & 0 & 0 & 1 \end{array}\right)\;.$$
Interestingly, the first two conservation laws, the momentum and energy (17), see also (13), (14), survive the stochastic deformation, while, as indicated by computer algebra, the third conservation law $\left(q^{\left(3\right)},p^{\left(3\right)}\right)$ is *broken*. Hence, the deformation of the matter–field automaton likely no longer posses nontrivial ergodicity breaking.

## 3. Deformed (Quantized) Hardpoint Lattice Gas

Nevertheless, we show briefly in this section how continuous deformations of some other reversible deterministic cellular automata can lead to interesting non-ergodic dynamics. For this purpose, we use the so-called charged hardpoint lattice gas [10] (also referred to as XXC model in [16] since it is a deterministic limit of models introduced and studied by Maassarani [17]).

In comparison to the matter–field automaton, the model has a richer local configuration space, i.e., three states per site, and a simpler local interaction map, applying only to a pair of neighbouring sites instead of three. In physical terms, the model describes the dynamics of free point particles which move with speeds ±1, with initial positions placed at commensurate (multiples of integer) coordinates, but which may carry an internal (say binary) degree of freedom (charge). Denoting the three local states as s∈{0,+,−}, with 0 referred to as a vacancy, the local reversible and deterministic rule, which we apply sequentially to pairs (2x,2x+1) and (2x+1,2x), reads
(24)(00)⟷(00),(0α)⟷(α0),(αβ)⟷(αβ),α,β∈{+,−}.
The model allows for a remarkable set of exact results on equilibrium and non-equilibrium dynamics, ranging from dynamical correlations in equilibrium and the transport coefficients [10], to an analytic *matrix-product-state* description of time-dependent quenches and non-equilibrium steady states in a boundary driven setup [11], to a recent exact solution of full counting statistics [12]. One may wonder if the model still allows for some degree of solvability, if hard-core collision is relaxed to a general stochastic, or unitary scattering
(25)$$\left(+-\right)⟶u_{11}\left(+-\right)+u_{12}\left(-+\right),\;\left(-+\right)⟶u_{21}\left(+-\right)+u_{22}\left(-+\right),$$
where *u* may be a 2×2 stochastic, or unitary matrix. In the latter case, we think of the quantum lattice gas model and quantum superposition states as elaborated precisely below. We note that this system is closely related to a semiclassically quantized sine-Gordon model, recently studied in [18].

Consider a local Hilbert space $H_{1}=C^{3}$, with states ${|s\rangle }_{s=0,+,-}$. We refer to states |±〉 as *charged particles* and we denote them by the Greek index ${|\alpha \rangle }_{\alpha =\pm }$, and to the state |0〉 as a *vacancy*. Let us define a local propagator over $H_{1}⊗H_{1}$ as
(26)$$\begin{array}{ccc} U[u] & = & |00\rangle \langle 00|+\sum_{\alpha \in \{\pm \}}\left(|0\alpha \rangle \langle \alpha 0|+|\alpha 0\rangle \langle 0\alpha |+|\alpha \alpha \rangle \langle \alpha \alpha |\right)+\\ & + & u_{11}|+-\rangle \langle +-|+u_{12}|+-\rangle \langle -+|+u_{21}|-+\rangle \langle +-|+u_{22}|-+\rangle \langle -+|, \end{array}$$
where *u* can be in principle any invertible matrix, u∈GL(2), for the properties that will be discussed below, while for quantum physics applications we will think of *u* as a unitary mixing matrix. Such U[u] satisfies two remarkable identities, valid for any *u*: (27)$$\begin{array}{ccc} & & U_{12}\left[u\right]U_{23}\left[1\right]U_{12}\left[u^{-1}\right]=U_{23}\left[u^{-1}\right]U_{12}\left[1\right]U_{23}\left[u\right], \end{array}$$(28)$$\begin{array}{ccc} & & U\left[u\right]U\left[u^{-1}\right]=1, \end{array}$$
which can be checked by straightforward computation. Note that identity (27) bears some similarity to the braid group form of the Yang–Baxter equation if *u* is considered as a non-abelian spectral parameter. Specifically, U[1] obeys the braid relation—it constitutes the so-called Yang–Baxter map—and provides a deterministic update rule for the charged hardpoint lattice gas. However, U[u] does not have the full Yang–Baxter property, but ust enough to render the existence of an infinite set of conserved local operators, as shown below.

For simplicity we can now assume a finite system of an even number of sites *L* and define its many-body Hilbert space as a tensor product $H=H_{1}^{⊗L}$. Then, we define a locality-preserving discrete time dynamical system (a quantum cellular automaton) over it, in terms of a brickwork circuit, i.e.,
(29)$$U=U_{o}U_{e},\;U_{e}=\prod_{x=1}^{L/2}U_{2x-1,2x}\left[u\right],\;U_{o}=\prod_{x=1}^{L/2}U_{2x,2x+1}\left[u\right],\;L+1\equiv 1,$$
and write the evolution of observables as
(30)$$a^{2t}=U_{e}a^{2t-1}U_{e}^{-1},\;a^{2t+1}=U_{o}a^{2t}U_{o}^{-1}.$$
Let us define the following pair of local operators (note periodic boundary conditions if needed):(31)$$\begin{array}{ccc} g_{x}^{\left(+\right)} & = & U_{x,x+1}\left[u\right]U_{x-1,x}U\left[1\right]U_{x,x+1}\left[u^{-1}\right],\\ g_{x}^{\left(-\right)} & = & U_{x-1,x}\left[u\right]U_{x,x+1}U\left[1\right]U_{x-1,x}\left[u^{-1}\right]. \end{array}$$
Observing identities (27) and (28), we can straightforwardly derive a few remarkable properties of these operators: (32)$$\begin{array}{ccc} U_{e}g_{2x}^{\left(+\right)}U_{e}^{-1} & = & g_{2x+1}^{\left(+\right)},\;U_{o}g_{2x-1}^{\left(+\right)}U_{o}^{-1}=g_{2x}^{\left(+\right)}, \end{array}$$(33)$$\begin{array}{ccc} U_{e}g_{2x}^{\left(-\right)}U_{e}^{-1} & = & g_{2x-1}^{\left(-\right)},\;U_{o}g_{2x+1}^{\left(-\right)}U_{o}^{-1}=g_{2x}^{\left(-\right)}, \end{array}$$
which justify the term glider operators [19] (see Figure 7 for a simple graphical proof of the properties (32) and (33) in terms of quantum circuit representation). We note that our gliders would be trivial $g_{x}^{\left(\pm \right)}=1$ for *integrable trotterization* [20] where U(1)=1.

These are particular kinds of quantum conservation laws, where the formal charge and current densities are identical. Consequently, we can build products of such operators, separated by at least two pairs of sites, which again behave in the same way
(34)$$g_{x_{1},x_{2}\ldots x_{m}}^{\left(\pm \right)}=\prod_{n=1}^{m}g_{2x_{n}}^{\left(\pm \right)},\;Ug_{x_{1},x_{2}\ldots x_{m}}^{\left(\pm \right)}U^{-1}=g_{x_{1}\pm 1,x_{2}\pm 1\ldots x_{m}\pm 1}^{\left(\pm \right)}$$
under constraints $x_{j+1}-x_{j}\geq 2$. We thus have a large set of extensive charges
(35)$$Q_{r_{1},r_{2}\ldots r_{m}}^{\left(\pm \right)}=\sum_{x=1}^{L/2}g_{x,x+r_{1}\ldots x+r_{m}}^{\left(\pm \right)}\;,\;r_{1}\geq 2,\;r_{m+1}-r_{m}\geq 2,\;Q_{\underline{r}}^{\left(\pm \right)}\equiv UQ_{\underline{r}}^{\left(\pm \right)}U^{-1}.$$
These conserved operators concisely encapsulate the non-ergodic properties of the deformed hardpoint lattice gas. The non-ergodicity should not be considered surprising, as the pattern of left and right movers is evolving regularly according to ballistic (free) dynamics. However, the internal (charge) degrees of freedom undergo interacting dynamics, which is nontrivially constrained by conserving $Q_{\underline{r}}^{\pm }$.

The fact that the number of distinct glider charges $Q_{\underline{r}}^{\left(\pm \right)}$ increases exponentially with the size of support, say *ℓ* such that all $r_{j}\leq \ell$, is reminiscent of superintegrable quantum cellular automata [13,15,16], yet the absence of an exact Yang–Baxter property suggests that the dynamical system is not completely integrable (i.e., there should be no Bethe ansatz!?). We are thus suggesting a new class of non-ergodic many-body quantum dynamics.

## 4. Conclusions

The dynamical systems introduced and discussed in this short contribution have been constructed for the sole amusement of the author. Yet, we hope these constructions may at some point find some nontrivial applications in the context of non-equilibrium statistical mechanics. Specifically, one of the long-standing key goals in the field is to derive irreversible macroscopic transport laws, such as Fick’s law of diffusion or Fourier’s law of heat transport, from microscopic reversible equations of motion. This question has also been among the main research focuses of Giulio Casati [21], as well as stimulating the beginning of my research career [22], and some of our joint work investigated Fourier’s law in systems very similar to the models studied here: hardpoint colliding masses in one dimension [23].

The coexistence of diffusive and ballistic transport has been rigorously demonstrated in the Rule 54 automaton (reviewed in ref. [13], see also ref. [24]), and one may hope that such results can be extended to the matter–field automaton, which seems richer and perhaps more generic (for example, Rule 54 dynamics is superintegrable with an exponentially growing number of conserved local charges, while the matter–field automaton is perhaps at most integrable, as it has much rarer nontrivial local charges).

## Figures and Tables

**Figure 1 entropy-25-00739-f001:**
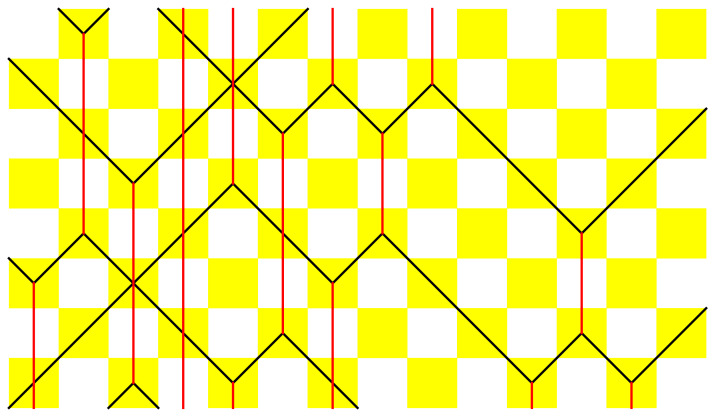
Example of an automaton’s trajectory (time runs in the vertical direction, say upwards). Yellow squares denote application of the local rule, or equivalently, of the three-bit permutation gate *Y*. Black/red lines denote the worldlines of matter/field quasiparticles.

**Figure 2 entropy-25-00739-f002:**
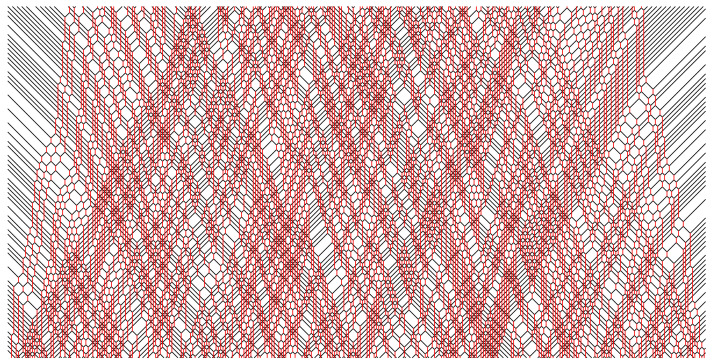
Example of a typical trajectory in a maximum entropy initial state, where $s_{x}^{t=0}$ are sampled randomly and uniformly in {0,1}, for a section of 0≤x≤600 and duration $t_{\max}=150$, $0\leq t\leq t_{\max}$. Note the evaporation of matter particles from the sides, and reduction in the field intensity, which is due to initial vacuum in the complement region, $s_{x^{\prime }}^{t=0}=0$ for $x^{\prime }<0$ or $x^{\prime }>600$.

**Figure 3 entropy-25-00739-f003:**
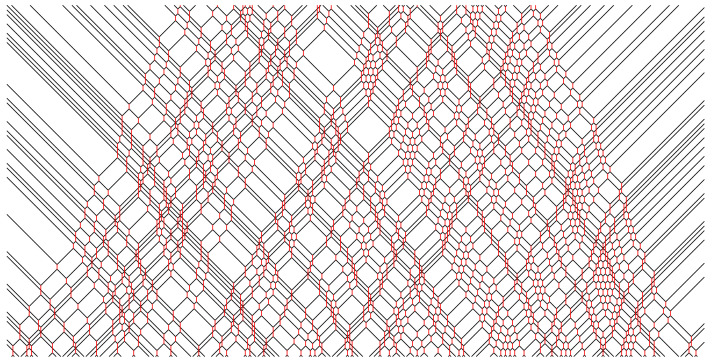
Example of a typical trajectory from the initial state where the matter is in a maximal entropy state, i.e., $s_{2x}$ are sampled randomly and uniformly for 0≤2x≤600, and there is no field, i.e., $s_{2x+1}\equiv 0$, of duration $t_{\max}=150$, $0\leq t\leq t_{\max}$.

**Figure 4 entropy-25-00739-f004:**
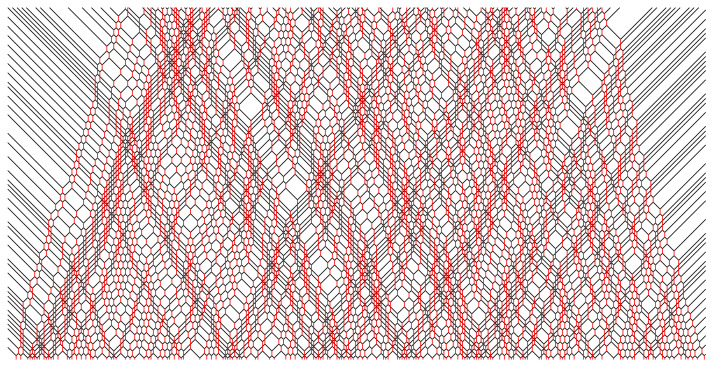
Example of a typical trajectory from the initial state where the field is in a maximal entropy state, i.e., $s_{2x+1}$ are sampled randomly and uniformly for 0≤2x+1≤600, and there is no matter, i.e., $s_{2x}\equiv 0$, of duration $t_{\max}=150$, $0\leq t\leq t_{\max}$.

**Figure 5 entropy-25-00739-f005:**
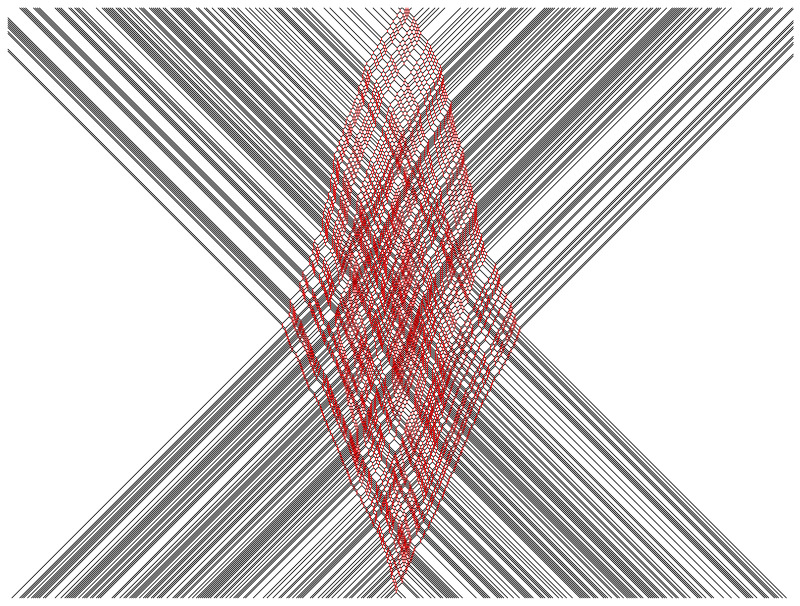
Head-on collision of a random (maximum entropy) sample of left movers with a random sample of right movers. The total width of each initial stream of left/right movers is 800 and the duration is $t_{\max}=600$.

**Figure 6 entropy-25-00739-f006:**
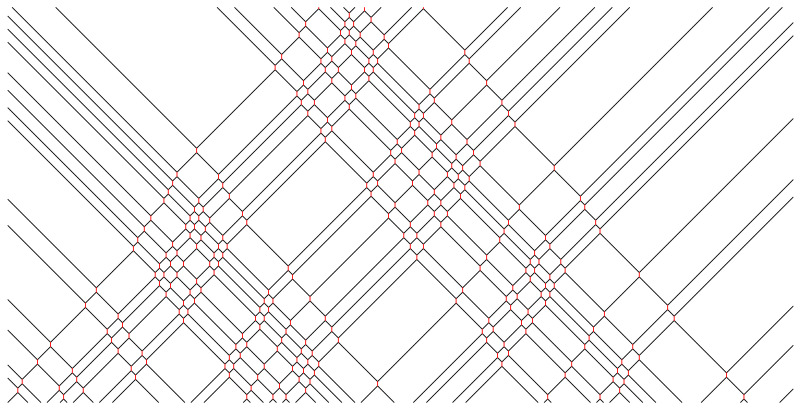
Trajectory starting from a random initial condition where only $s_{6x}$ (0≤x≤180) are chosen at random, uniform in {0,1}, while all other $s_{x^{\prime }}=0$. Such a trajectory features only maximally short-lived field particles, which provide ‘unit’ phase shifts for left/right movers upon scattering, and corresponds to dynamics of negative-length hard-rods, such as, e.g., in the Rule 54 system [13].

**Figure 7 entropy-25-00739-f007:**
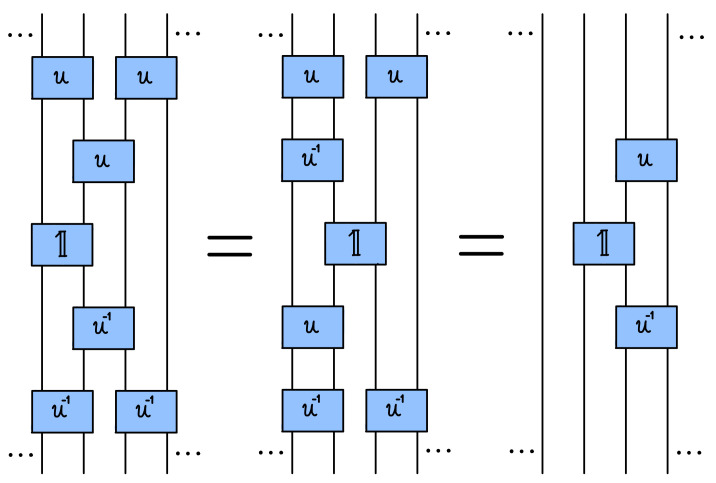
Diagrammatic proof of the glider property (32) in terms of equivalence of quantum circuits, where gates U[u],U[1], and $U[u^{-1}]$ are indicated by blue boxes. The proof of (33) is analogous (via left–right reflection of the above diagrams).

## Data Availability

All data shown or used in this paper have been generated by *Mathematica* encoding the procedures explicitly described above.

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
