# Peer review of "On Two Non-Ergodic Reversible Cellular Automata, One Classical, the Other Quantum†"

_entropy, 2023, doi:10.3390/e25050739_

Round 1

Reviewer 1 Report

The draft is of superb quality, introducing two interesting non-ergodic cellular automata that have not been reported previously. The two non-ergodic cellular automata are novel and they might be useful deciphering some mysteries in the field of integrable/non-ergodic quantum circuits, notably the integrability of deformed rule 54 quantum cellular automata. The simplicity and elegance of the two models seem to imply more general integrable structure, leading toward exact results in out-of-equilibrium properties. The "glider equation" seems to be novel and could potentially lead to new development in the integrable structure of quantum many-body systems.

The presentation of the paper is very clear and the English written flawless. The only comments that I have regarding the draft are about few typos that can be fixed easily.

1-- There is a typo in Eq. (10), it should read

$$

U_e q - U_o^{-1} q + \Pi p - \Pi^{-1} p = 0 .

$$

Similarly, for the gauge transformation Eq. (21), there's a typo

$$

p \rightarrow p - U_o a + U_e^{-1} a .

$$

2-- About the energy density and current Eq. (14-15): it can be written as 2 parts, combining both we get the second equation of Eq. (16), which implies an additional local conserved charges, namely the original one in Eq. 

$$

q_x^{(2)} = [ \dot ]_{4x+1} + [ \dot ]_{4x+2} + [ \dot ]_{4x+3} 

$$

$$

p_x^{(2)} = - [ \dot ]_{4x+2}

$$

and another pair

$$

q_x^{(2)\prime} = [ \dot ]_{4x-1} + [ \dot ]_{4x} + [ \dot ]_{4x+1} 

$$

$$

p_x^{(2)\prime} = + [ \dot ]_{4x}

$$

both satisfying continuity equation (10). And $q_x^{(2)\prime} = \Pi^{-1} q_x^{(2)} $.

Two conserved quantities are

$$

E_1 = \sum_{x=0}^{L/4-1} [ \dot ]_{4x+1} + [ \dot ]_{4x+2} + [ \dot ]_{4x+3} ,

$$

and 

$$

E_2 = \sum_{x=0}^{L/4-1} [ \dot ]_{4x-1} + [ \dot ]_{4x} + [ \dot ]_{4x+1} .

$$

Together they combine into the "energy" $E$ (or particle number) in second equation of (16)

$E_1 + E_2 = E $.

It would be great if the author could clarify this around Eq. (16).

3-- I wonder if the "glider equation" Eq.(26) could be baxterised into a dynamical version of Yang-Baxter equation. I am not an expert in the field of dynamical Yang-Baxter equation and I understand that the answer to this question might be beyond the scope of the current draft. I wonder if the author would have some ideas to this question.

In conclusion, I think the paper is nicely written and it can be published once the author fixes the minor typos mentioned above.

Author Response

I thank the referee for her/his careful reading of the manuscript. I have slightly revised it according to suggestions. Specifically, I have clarified the relationship between pair of observables (P,E) and (Q^1,Q^2), which involve some subtlety related to shift-invariance (which is now stated more explicitly).

Reviewer 2 Report

In this paper two `simple' models --- as cellular automata, are proposed and studied, for possible usefulness in the study of many body systems. I think that this exploring is good at least in view of two points: (i) cellular automata is among one of the simplest models available as far as I know, and (ii) study of many-body systems is one of the hardest tasks in physics. For this reason, I would like to recommend publication of the paper. 

Author Response

I thank the referee for positive evaluation of the manuscript